# Characterization of Bond Fracture in Discrete Groove Wear of Cageless Ball Bearings

**DOI:** 10.3390/ma15196711

**Published:** 2022-09-27

**Authors:** Yanling Zhao, Yuan Jin, Chengyi Pan, Chuanwang Wu, Xueyu Yuan, Gang Zhou, Wenguang Han

**Affiliations:** Key Laboratory of Advanced Manufacturing and Intelligent Technology, Ministry of Education, Harbin University of Science and Technology, Harbin 150080, China

**Keywords:** cageless ball bearings, discrete groove wear, bond fracture, particle shedding, discrete element method

## Abstract

Cageless ball bearings with discrete grooves in the outer raceway enable the dispersion of rolling elements. Once worn, the discrete groove can cause the rolling element to discretely fail. This paper presents the discrete element method to investigate the wear of discrete grooves in cageless bearings from the standpoint of bond fracture. In conjunction with the structural characteristics of bearing races with discrete slots, we propose a hexagonal close-spaced spherical particle arrangement, in which the discrete slots are discretized into particles of the same size that are connected by bonds. The contact model and contact force equation between the rolling elements and the aggregate elements are established, and the external force on the aggregate elements is calculated. Under the influence of an external force and the arrangement of particles in the aggregate element, the internal force transfer equation of different layers and different particles is derived, and the internal force of the particles in the aggregate unit is calculated. In accordance with Hertz–Mindline theory, the bonding model of discrete groove particles is established, the size of the particle shedding cohesive force during bond fracture is determined, and the wear degree of discrete grooves is characterized by comparing the cohesive force and internal force. Numerical solutions and wear tests are combined. Bond fracture can accurately characterize the wear of discrete grooves. This approach offers theoretical guidance for cageless bearing design.

## 1. Introduction

Collisions among rolling elements in cageless bearings occur due to the absence of a cage. To achieve the stability of cageless bearings, Zhao et al. proposed a method of setting a reduced collision groove in the outer raceway to make the rolling elements discrete [1,2]. However, the constant contact between the rolling elements and the discrete groove will lead to wear of the discrete groove, which will affect the discrete rolling elements [3]. It is therefore important to investigate the wear of discrete grooves in cageless ball bearings.

For bearing wear, most scholars use the Archard model [4,5,6]; however, these scholars have not analyzed the model from a material point of view. Some scholars have investigated the mechanical aspects of wear models for materials. Holm portrays wear as an atomic process in which the material being removed is treated as a single atom and the probability of both atoms being pulled out of their parent surface is constant for each contact between two surface atoms in the contact area. An empirical linear wear relationship between the amount of wear and the applied load was obtained from the study [7]. To verify the validity of the theory proposed by Holm, Ramin et al. used molecular dynamics to test the basic assumptions of the Archard wear model and found that the Archard wear model was consistent with the wear pattern obtained from molecular dynamics [8]. Brian et al. developed a wear model from the distribution of indentations on surface contact areas, which incorporates internal material forces and takes into account microcontact loads, the angle of the microcontact particles and the modulus to provide a basis for analyzing material fracture thresholds [9]. Yang et al. proposed an internal mechanics method and applied it to estimate the effective wear of nonlinear inhomogeneous materials with complex intrinsic relationships; this method is useful for understanding and designing the tribological properties of nonlinear inhomogeneous materials [10]. Some scholars have considered the internal structure of a material and developed a mechanical model of the material’s interior. Foster monitored the internal organization of the constituent phases in the internal structure of bearing steels during the tensile process, compared it with the internal organization of standard quenching, and obtained the mechanical behavior of the different internal organization components in relation to the quenching process parameters [11]. Matti et al. investigated the internal force structure of the sliding contact of wear-resistant, high manganese steel abrasives, established a micromechanical model of high manganese steel, conducted a numerical study of single contact wear of high manganese steel and analyzed the influence of grain orientation and internal structural characteristics on wear [12].

A very promising tool is the discrete element method proposed by Cundall and Strack, which proposes a bonding model from an internal material perspective for discretizing a continuum into particles and for connecting the individual particles with bonds [13]. Several scholars conducted wear studies based on the properties of bonds. Machado et al. used the discrete element method to build a two-dimensional model of a bearing, which considered the inner and outer rings as a collection of particles, simulated the load distribution and stress distribution of the outer ring of the bearing under different operating conditions, and verified the results with each other in the finite element simulation, providing a basis for the model to be utilized in the study of wear [14]. Ma et al. employed discrete element software (PFC2D, ITASCA, Minneapolis, MN, USA) to simulate the frictional wear process of a copper-based, plain bearing material against 45 steel, where the amount of wear is represented by the amount of particle shedding due to bond fracture [15]. Zhao et al. established discrete element models based on the discrete element method for ceramic and workpiece materials to simulate the evolution of tool crack generation, expansion and material spalling during machining of 45 steel with ceramic tools and to analyze the effect of different parameters on tool wear [16]. Zhang et al. used discrete element software to establish the step, convexity and scale arrangement structures of wear surfaces and their abrasive wear systems to observe light injuries and shedding of abrasive wear behavior through qualitative analysis of abrasive grain morphology, contact bond fields and contact force chains [17]. Based on the bonding model, Horabik et al. employed the discrete element method to simulate the fragmentation process of starch agglomerates and to analyze the microscopic mechanism of agglomerate compression and bond breakage strength [18]. The discrete element method was improved by Feng et al. by establishing a fracture criterion for steel through the fracture criterion for bonds, effectively solving the problems of large deformation, plasticity and fracture of beam structures [19]. Horabik et al. modeled agglomerate fragmentation using the discrete element method, replacing adhesion in relaxed particle contact with bonding, and providing greater insight into the mechanism of interparticle bonding [20]. Shilko et al. used the discrete element method to describe the fracture process as a process from linked state to unlinked state, and on this basis, the fracture criterion of bonding bond was established [21]. Smolin et al. established a criterion for bond fracture-based energy, explicitly modeled the formation and evolution of a microscale third body as well as investigating the influence of material properties (and structure) on the mechanisms of wear and the coefficient of friction [22].

In densely arranged particulate matter, the particles have minimal space to freely move, and gravity or external loads squeeze and deform each other, transferring external forces and forming force chains. Based on the strict particle contact theory, the discrete element method is able to analyze contact forces and force chains in three-dimensional particulate matter [23]. Fu et al. used discrete element modeling to simulate biaxial compression tests and analyzed the evolution of the force chain distribution throughout the compression process, providing direct micromechanical evidence for the macroscopic behaviors of granular materials [24]. Zhang et al. investigated the lengths of force chains in high-voltage, DC transmission by the discrete element method and quantitative analysis methods and discussed the relationship between the lengths of force chains and other characteristics of the force chains [25]. Meng et al. used the discrete element method to simulate the evolution of force chains in shear particle friction systems and considered the effects of pressure loading and shear velocity to elucidate the important relationship between force chains and particle flow states [26]. Liu et al. employed the discrete element method to simulate triaxial compression tests on cobbles, analyzed the evolution of force chains during compression and discussed the instability of cobble tunnels on this basis [27]. Some scholars have considered the quantitative analysis of force chains. Jiang et al. investigated the force transfer law of an ordered, symmetrically arranged, two-dimensional particle pile, applied stochastic theory to establish a force transfer model [28] and then carried out a probabilistic analysis of quantitative experiments on force transfer in a particle pile under concentrated loading to obtain a stochastic model of force chains from generation to gradual disappearance [29]. The distribution of bottom surface forces in hexagonal dense-row accumulations of spherical particles under concentrated forces was quantitatively investigated by Miao et al. [30]. Zhang et al. simplified the two-dimensional particle stacking to a lattice point system and established a force transfer equation to theoretically derive the relationship between the force distribution at the bottom of the stack and the force distribution at the top of the stack [31]. Sun et al. applied Saint Venant’s principle to establish a force transfer model for a two-dimensional particle pile and determined the stress distribution in the jaws for different crushing particle sizes [32].

Current research on the internal forces of materials, especially the application of the discrete element method to bearing wear, lacks a quantitative analysis of force chains, and no scholar has applied the discrete element method to the study of bearing wear. Therefore, this paper investigates the wear of discrete grooves in cageless ball bearings based on the discrete element method. First, the discrete groove is discretized into particles connected by bonds, and the bond fracture condition is determined. Second, some of the particles are formed into a collection unit, and the external force applied to the collection unit is determined. Last, the internal force distribution of the particle system under the action of the external force is analyzed, the internal force applied to each particle is quantified, and the bond fracture amount is determined according to the bond fracture condition to obtain the particle shedding amount, that is, bearing wear.

## 2. Methods

### 2.1. DEM Model of a Cageless Ball Bearing

In this study, using the discrete element method, the outer ring of a cage free ball bearing is discretized into particles of the same size, and the particles are bonded to form a complete bearing outer ring, as shown in Figure 1a. The inner raceway speed is *ω_i_*, and the circumferential span angle of discrete groove is *θ*. Due to the large difference in size between the particles and the rolling element and the large number of particles in the inner and outer rings, the particles in contact with the rolling element are combined to form a collective element. The force transfer analysis of the particles is then carried out, with each rolling element forming an aggregate element *V* at the contact with the particles. As shown in Figure 1b, the rolling element rotation speed is *ω_b_*, and the common angle speed of the rolling element is *ω_m_*; the rolling element is in contact with the two aggregate elements at the reduced bumper groove. Figure 1c shows an enlarged view of the aggregate element, where the red lines represent the previously described bonds.

#### 2.1.1. Contact Model of the Rolling Element and Aggregate Element

In this study, the Hertz-Mindlin contact model was selected for the rolling element to aggregate element contact. A virtual overlap δ occurs when contact occurs between the rolling element and the aggregate element, and the contact is made without regarding deformation. Figure 1a shows a model of a cageless ball bearing with a discrete groove in contact with the outer aggregate element. The contact force resulting from the contact between the rolling element and the outer ring is
(1)$$F_{i}=\sum F_{nbV}+F_{tbV},$$
where ***F****_nbV_* is the normal contact force and ***F**_tbV_* is the tangential contact force. In the discrete element method, particles are modeled using soft spheres. In contact, only the normal overlap of the two particles and the tangential displacement are considered. A model of the rolling element in contact with the aggregate element is shown in Figure 1d. In Figure 1d, *R_b_* is the radius of the rolling element, and *R_V_* is the length of the short semi axis of the ellipse. When the two elements make contact, a dummy overlap *δ* appears on the normal side. *δ* affects the magnitude of the contact force, and the normal contact force is
(2)$$F_{nbV}=\left(-k_{n}\delta^{3/2}-c_{n}Gn\right)n,$$
where *k_n_* is the normal elasticity coefficient, *c_n_* is the normal damping coefficient, and ***G*** is the velocity of the rolling element relative to the aggregate element *V*. The tangential contact force is
(3)$$F_{tbV}=-\mu \left|F_{nbV}\right|t,$$
where *μ* is the coefficient of static friction, and ***t*** is the tangential element vector. The tangential force is disregarded, as it has a minimal effect on the following study [23]. The normal elasticity coefficient *k_n_* is
(4)$$k_{n}=\frac{4}{3}{\left(\left(1-\upsilon_{b}^{2}\right)/E_{b}+\left(1-\upsilon_{V}^{2}\right)/E_{V}\right)}^{-1}{\left(\left(\left(D_{w}/2\right)+R_{V}\right)/\left(D_{w}/2\right)R_{V}\right)}^{-1/2},$$
where *v_b_* is the rotational speed of the rolling element, *v_V_* is the aggregate element velocity, *E_b_* is the modulus of elasticity of the rolling element, *E_V_* is the modulus of elasticity of the aggregate element, *D_w_* is the rolling element diameter, and *R_V_* is the radius of the aggregate element. The equation for the normal phase damping factor is
(5)$$c_{n}=-2\ln e\sqrt{k_{n}m_{b}m_{V}/\left(m_{b}+m_{V}\right)}/\sqrt{\pi^{2}+\ln^{2}e},$$
where *m_b_* and *m_V_* are the mass of the rolling element and the mass of the aggregate element, respectively, and *e* is the collision recovery factor.

According to the above analysis of the rolling element and aggregate element contact model, combined with Formulas (2)–(5), the contact force of the rolling element and aggregate element is
(6)$$F_{i}=\left(\begin{array}{l} -\frac{4}{3}{\left(\left(1-\upsilon_{b}^{2}\right)/E_{b}+\left(1-\upsilon_{V}^{2}\right)/E_{V}\right)}^{-1}{\left(\left(R_{b}+R_{V}\right)/R_{b}R_{V}\right)}^{-1/2}\delta^{3/2}-\\ \left(-2\ln e\sqrt{k_{n}m_{b}m_{V}/\left(m_{b}+m_{V}\right)}\left(v_{b}-v_{V}\right)/\sqrt{\pi^{2}+\ln^{2}e}\right)n \end{array}\right)n,$$

#### 2.1.2. Model for Bond Fracture of Raceway Particles

As described in the previous section, the outer raceway of the bearing is discretized into particles of the same diameter, which are interconnected by bonds. To ensure that the nature of the discrete bearing does not change and in accordance with the particular shape of the bearing raceway, the particles of the bearing retaining edge are determined to exhibit a simple cubic arrangement, as shown in Figure 2a. The particles in the conventional raceway and at the discrete groove show a hexagonal close-packed (hcp) arrangement, as shown in Figure 2b.

The contact between the raceway particles is also consistent with the Hertz-Mindlin contact model mentioned in the previous section, where the tangential and normal spring stiffnesses need to be determined. According to the law of conservation of energy, the strain energy of the simple cubic and hcp arrangements are
(7)$$U_{si}=\frac{1}{V}\sum_{j=1}^{6}\frac{1}{2}\left[\frac{1}{2}kn_{sij}{\left(u_{sn,j}-u_{sn,i}\right)}^{2}+\frac{1}{2}ks_{sij}{\left(u_{ss,j}-u_{ss,i}\right)}^{2}\right],$$
(8)$$U_{hi}=\frac{1}{V}\sum_{j=1}^{12}\frac{1}{2}\left[\frac{1}{2}kn_{hij}{\left(u_{hn,j}-u_{hn,i}\right)}^{2}+\frac{1}{2}ks_{hij}{\left(u_{hs,j}-u_{hs,i}\right)}^{2}\right],$$
where *V* is the equivalent volume; *kn_sij_* and *kn_hij_* are the spring normal stiffness between particle *i* and particle *j* in the simple cubic arrangement and hcp arrangement, respectively; *ks_sij_* and *ks_hij_* are the spring tangential stiffness between particle *i* and particle *j* in the simple cubic arrangement and hcp arrangement, respectively; *u_sn,j_* and *u_hn,j_* are the particle *j* normal phase displacements in the two arrangements; *u_sn,i_* and *u_hn,i_* are the normal phase displacements of particle *i* in the two arrangements; *u_ss,j_* and *u_hs,j_* are the tangential phase displacements of particle *j* in the two arrangements; and *u_ss,i_* and *u_hs,i_* are the tangential phase displacements of particle *j* in the two arrangements.

Through coordinate transformation and physical equations in the form of energy in elastodynamics, combining Equations (7) and (8), the stiffness of the two arrangements is expressed as
(9)$$\begin{array}{l} kn_{sij}=\frac{2VE}{5l_{0}{}^{2}(1-2\nu )}\\ ks_{sij}=\frac{2VE(4\nu -1)}{5l_{0}{}^{2}(2\nu -1)(\nu +1)} \end{array}$$
(10)$$\begin{array}{l} kn_{hij}=\frac{3VE}{4l_{0}{}^{2}(1-2\nu )}\\ ks_{hij}=\frac{3VE(4\nu -1)}{4l_{0}{}^{2}(2\nu -1)} \end{array}$$
where *E* is the elastic modulus of the bearing steel, *ν* is Poisson’s ratio of the bearing steel, and *l*_0_ is the initial distance between the spherical centers of the two particles.

The particle center of the mass distance is less than the particle radius, and there are contact and bonding forces between two particles when the forces in the HMB model are calculated according to the Hertz–Mindlin contact model. Thus, combining Equations (2), (9) and (10) yields the bonding force between two particles in the two different arrangements as
(11)$$Fn_{s}=kn_{sij}\delta^{3/2},$$
(12)$$Fn_{h}=kn_{hij}\delta^{3/2},$$

When the indirect contact force of the two particles is greater than the bonding force of the bond between these particles, the bonding key breaks, that is, the bearing is worn.

### 2.2. Wear Study of Ball Bearings without a Cage

This paper mainly analyzes the wear characteristics of cageless ball bearings from the perspective of internal forces. In the previous section, the bonding force of the bond between two raceway particles and the bond fracture conditions were determined. In this section, external forces are converted to internal forces, the force distribution characteristics in the particle system under the action of external loads are analyzed, and the bond fracture conditions in the discrete element model of the bearing outer ring are obtained.

#### 2.2.1. Analysis of External Forces in Particle Systems

To analyze the force distribution characteristics in the particle system, it is necessary to determine the external force on the aggregate element under an external load, that is, the contact force between the rolling element and the aggregate element. According to Equation (6), the main influencing factors of the contact force between the rolling element and the aggregate element are the overlap *δ* and the relative speeds of the rolling element and aggregate element.

The contact overlap *δ* between the rolling element and the aggregate element is determined using Hertzian theory:(13)$$\delta ={\left(9P^{2}/16RE^{*2}\right)}^{1/3},$$
where *P* is the total load between the rolling element and the aggregate element, *E** is the equivalent elastic modulus and *R* is the relative curvature:(14)$$R=\frac{1}{R_{b}+R_{V}},$$

The force analysis reveals that in the bearing region, the total load *P* is composed of the rolling element itself, gravity *G*, and centrifugal force *F_c_*, whereas the rolling element is composed of the radial load component *Q_ψ_*. According to the literature [33], the formula for *Q_ψ_* is expressed as
(15)$$Q_{\psi }=Q_{max}{\left[1-\frac{1}{2T}\left(1-\cos\psi \right)\right]}^{\frac{3}{2}},$$
where *ψ* is the angle between the rolling element and the vertical direction, *Q_max_* is the maximum rolling element load, and *T* is the load distribution factor, which is related to the radial clearance. The radial clearance of the bearing in this paper is 0, so *T* = 0. Formula (15) shows that different rolling elements are subjected to different load components *Q_ψ_*, which are related to their position angle. Thus, the force analysis of the cageless ball bearing is carried out as shown in Figure 3.

As shown in Figure 3, for the radial load *Q*, the upper half of the rolling element due to the inner ring is oriented downward and is not bearing, but the lower half of the rolling element will bear the load. Thus, in the upper half of the rolling element in the non-bearing region, the total load between the rolling elements and the outer raceway is expressed by the sum of gravity and centrifugal force. Due to the rolling element in the groove when the mass of the touch down and the inner raceway do not contact, the rolling element does not carry, as it is only affected by gravity *G* and centrifugal force *F_cg_*; at this time, the total load between the rolling element and the touch groove is only the rolling element’s own gravity and centrifugal force.

According to the above analysis, the contact deformation between the rolling element and the aggregate element in the conventional raceway bearing region is
(16)$$\delta ={\left(\frac{9{\left(G\cos\psi +Q_{max}{\left(\cos\psi \right)}^{3/2}+m\omega_{m}^{2}D_{m}/2\right)}^{2}}{16RE^{*2}}\right)}^{1/3},$$
where *ω_m_* is the common angular velocity of the rolling element, and *D_m_* is the pitch diameter. The contact deformation between the rolling element and the aggregate element in the non-bearing region of the conventional raceway is
(17)$$\delta ={\left(\frac{9{\left(G\cos\psi +m\omega_{m}^{2}D_{m}/2\right)}^{2}}{16RE^{*2}}\right)}^{1/3},$$

Since the contact between the rolling element and the discrete groove is a two-point contact, the rolling element will simultaneously make contact with both aggregate elements in the discrete groove. The deformation of the rolling element and aggregate element at the discrete groove is
(18)$$\delta ={\left(\frac{9{\left(\left(G\cos\psi +m\omega_{mg}{}^{2}\Delta r\right)\cos\left(\arcsin\left(2r_{o}\sin\left(\theta /2\right)/D_{w}\right)\right)\right)}^{2}}{16RE^{*2}}\right)}^{1/3},$$
where *ω_mg_* is the common angular velocity when the rolling body passes through the discrete groove, Δ*r* is the contact radius of the discrete groove, *r_o_* is the curvature radius of the outer raceway of the bearing, and *D_w_* is the diameter of the rolling element. Next, the rolling element is analyzed relative to the velocity of the aggregate element. Figure 4a shows a schematic of the relative velocity of the rolling element and aggregate element at the conventional raceway, with the relative velocity of the two being
(19)$$G_{bVo}=V_{b}-V_{Vo},$$
where *V_Vo_* is the aggregate element linear velocity and *V_b_* is the rolling element linear velocity. Figure 4b shows a schematic of the relative velocity of the rolling element and aggregate element at the deconfliction slot, with the relative velocity of the two being
(20)$$G_{bVo}{}^{\prime }=V_{b}{}^{\prime }-V_{Vo},$$
where *V_b_′* is the linear velocity of the rolling element as it passes through the discrete groove.

Since the direction of the relative velocity of the rolling element and aggregate element is perpendicular to the direction of the contact force, the damping force in the contact force equation is 0. Combining Equations (6), (16) and (19), we obtain the contact force between the rolling element and the aggregate element in the bearing region at the conventional raceway as
(21)$$F_{i}=\left(\begin{array}{l} -\frac{4}{3}{\left(\left(1-\upsilon_{b}^{2}\right)/E_{b}+\left(1-\upsilon_{V}^{2}\right)/E_{V}\right)}^{-1}{\left(\left(R_{b}+R_{V}\right)/R_{b}R_{V}\right)}^{-1/2}\\ {\left(\frac{9{\left(G\cos\psi +Q_{max}{\left(\cos\psi \right)}^{3/2}+m\omega_{m}^{2}D_{m}/2\right)}^{2}}{16RE^{*2}}\right)}^{1/2} \end{array}\right)n,$$

Combining Equations (6), (17) and (19), we obtain the rolling element and aggregate element of the raceway in the non-bearing region at the conventional raceway as
(22)$$F_{i}=\left(-\frac{4}{3}{\left(\frac{1-\upsilon_{b}^{2}}{E_{b}}+\frac{1-\upsilon_{V}^{2}}{E_{V}}\right)}^{-1}{\left(\frac{R_{b}+R_{V}}{R_{b}R_{V}}\right)}^{-1/2}{\left(\frac{9{\left(G\cos\psi +m\omega_{m}^{2}D_{m}/2\right)}^{2}}{16RE^{*2}}\right)}^{1/2}\right)n,$$

Combining Equations (5), (18) and (19), we obtain the contact force between the rolling element and the aggregate element at the discrete groove as
(23)$$F_{i}=\left(\begin{array}{l} -\frac{4}{3}{\left(\left(1-\upsilon_{b}^{2}\right)/E_{b}+\left(1-\upsilon_{V}^{2}\right)/E_{V}\right)}^{-1}{\left(\left(R_{b}+R_{V}\right)/R_{b}R_{V}\right)}^{-1/2}\\ {\left(\frac{9{\left(\left(G+m\omega_{mg}{}^{2}\Delta r\right)\cos\left(\arcsin\left(2r_{o}\sin\left(\theta_{2}/2\right)/D_{w}\right)\right)\right)}^{2}}{16RE^{*2}}\right)}^{1/2} \end{array}\right)n.$$

#### 2.2.2. Analysis of the Internal Force Transfer in a Raceway Particle System

In the previous section, the contact forces between the rolling element and the aggregate element at different positions in the raceway were obtained. In this section, the transfer characteristics of the internal forces in the particle system are analyzed.

A schematic of the contact between the rolling element and the discrete rear outer raceway is shown in Figure 5. For rolling elements in contact with *n_c_* particles, each contact is referred to as a microcontact [14]. For force transfer in space, the analysis is performed using an aggregate element containing *Np* particles rather than for individual particles. In this study, the aggregate element is ellipsoidal, and the aggregate element contains all particles in contact with the rolling element. As shown in Figure 5a, the rolling element rolls to different positions on the outer raceway and will make contact with different aggregate elements, forming different contact forces, which will then be distributed to the small particles in the aggregate units:(24)$$F_{i}=\sum_{i=1}^{n_{c}}\vec{q_{i}}\cdot \vec{n},$$
where *q_i_* is the internal force generated by each microcontact.

As shown in Figure 5b, the rolling element makes contact with the aggregate element, forming a contact ellipse whose area is equal to the sum of the contact circle areas of all particles in contact with the rolling element [14]:(25)$$S_{i}=\sum_{i=1}^{n_{c}}s_{i},$$

According to the literature [31], the pressure distribution within the contact ellipse is
(26)$$p\left(x,y\right)=p_{0}\sqrt{1-\frac{x^{2}}{a^{2}}-\frac{y^{2}}{b^{2}}},$$
where *p_0_* is the pressure at the center of the ellipse between the rolling element and the aggregate element, and *a* and *b* are the long semiaxis and short semiaxis, respectively, of the contact ellipse.

The distribution of internal forces within a contact ellipse consisting of microcontacts is
(27)$$q\left(x,y\right)=p\left(x,y\right){\left(d/2\right)}^{2},$$
where *d* is the particle size.

The above analysis gives the value of the internal force for each particle in contact with the rolling element on the raceway, which is also the force on the first layer of particles in the particle system. Therefore, the next step is to carry out an analysis of the internal force transfer in the particle system. According to the analysis in Section 2.1.2, the arrangement of the raceway particles is hcp, and therefore, force transfer analysis is carried out for the hcp structured particle system. The basic assumptions of the hcp internal force transfer model are presented as follows:The transmission of tangential forces between two particles is not considered;Since the lateral forces balance each other, only the transmission of the vertical component of the internal force is considered;The displacement and deformation of particles do not change the force transfer characteristics.

As shown in Figure 6, for the internal force transfer model of hcp stacking, there are m + 1 layers of hcp stacking particles in the raceway of the cageless ball bearing as a whole, 1–7 is the number of different particles in each layer. The transfer model applies stochastic theory and divides the transfer of forces into an equally distributed component and a fluctuating volume ε component [28]. In a particle stack in the hcp arrangement, each particle away from the boundary is subjected to a force transmitted by three neighboring particles in the upper layer and transmits the force to three neighboring particles in the next layer. The particle immediately adjacent to the boundary transmits the force to one of the neighboring particles in the next layer. By mathematical induction, the general formula for internal force transfer is expressed as
(28)$$F_{m,k}=\left[\begin{array}{ccc} F_{m-1,k_{1}} & F_{m-1,k_{2}} & F_{m-1,k_{3}} \end{array}\right]\left[\begin{array}{c} f_{m-1,k_{1}}\\ f_{m-1,k_{2}}\\ f_{m-1,k_{3}} \end{array}\right],$$
where *F_m,k_* is the internal force transmitted by the kth particle of the mth layer to the previous layer; *F_m_*_−1,*k*1_ is the internal force transmitted by the *k*_1_th particle of the (m − 1)th layer to the previous layer; *F_m_*_−1,*k*2_ is the internal force transmitted by the *k*_2_th particle of the (m − 1)th layer to the previous layer; and *F_m_*_−1,*k*3_ is the internal force transmitted by the *k*_3_th particle of the (m − 1)th layer to the previous layer. *f_m_*_−1,*k*1_, *f_m_*_−1,*k*2_ and *f_m_*_−1,*k*3_ are the transfer ratios of the three particles in layer m-1 to the kth particle in layer m, which are given by
(29)$$\begin{array}{l} f_{m-1,k_{1}}=\frac{1}{3}+a\varepsilon_{m-1,k}\\ f_{m-1,k_{2}}=\frac{1}{3}+b\varepsilon_{m-1,k}\\ f_{m-1,k_{3}}=\frac{1}{3}+c\varepsilon_{m-1,k} \end{array}$$
where *ε_m_*_−1,*k*_ is fluctuation quantities, *a*, *b* and *c* is fluctuation coefficient, and *a* + *b* + *c* = 0.

According to the literature [34], and due to the hcp arrangement, the particles in the middle region are subjected to the force of the upper three particles and transmit the force to the lower three particles, while the particles at the border are subjected to the force of only one particle in the upper layer and transmit the force to only one particle in the lower layer. Therefore, the transfer ratio in the middle region is 1/3, and the transfer ratio at the border is 1. In addition, the initial value in the internal force transfer general term equation is the internal force on each particle in the first layer, which is expressed as
(30)$$F_{1,k}=q_{k}.\sin\theta_{k},\;k=\;1,2,3\text{…}i$$
where *θ_k_* is the angle between the internal force on each particle in the first layer and the vertical direction.

## 3. Experimental

This study uses accelerated life testing of bearings. The experimental equipment is shown in Figure 7. The specific method is to change the material of the test bearing to 45 steel and then carry out the wear test of the bearing under variable speed and variable load conditions. The test scheme is shown in Table 1.

## 4. Results and Discussion

### 4.1. External Force

When the radial load of the bearing is 2000 N, considering the bearing at different speeds, Equations (21)–(23) are solved. Using the solution parameters in Table 2, rolling element and aggregate element contact force change law, the same equation is solved for different speeds, as shown in Figure 8.

When the position angle 90° < *ψ* < 270°, contact occurs between the rolling element and the aggregate element in the region of the conventional raceway. When the position angles 12° < *ψ* < 90° and 90° < *ψ* < 348°, contact occurs between the rolling element and the aggregate element in the conventional raceway bearing region. When the position angle *ψ* < 12° and *ψ* > 348°, contact occurs between the rolling element and the assembly element on the reduced bumper groove. As shown in Figure 8, the contact force between the rolling element and the aggregate element in different positions of the raceway increases with an increase in the inner ring speed. The contact force in the non-load bearing region under any speed is the smallest, as the rolling element is not subject to radial load at this time, and the aggregate element is only subject to centrifugal force and gravity. In the conventional raceway bearing region, the contact force is the largest, which is closer to the bearing vertical centerline position, and the greater the contact force, the larger the load between the rolling element and the aggregate element, due to an increase in the rolling element of the gravity component, resulting in increased contact force. The contact force between the rolling element and the aggregate element is minimized in the reduced contact groove, as the rolling element is not subject to the radial load as the center of mass drops in the reduced contact groove, and the contact radius of the rolling element decreases, resulting in a smaller centrifugal force. The contact between the rolling element and the aggregate element changes from one point to two points, that is, the rolling element will make contact with two aggregate elements. Therefore, the load between the rolling element and each aggregate element decreases, resulting in a smaller overlap and thus a smaller contact force.

### 4.2. Internal Force

This paper presents a numerical simulation of the internal forces in a granular system with an inner ring speed of 6000 rpm and a radial load of 2000 N as an example.

#### 4.2.1. Initial Force

The internal force distribution in the contact ellipse is solved according to the equation in Section 2.2.2. The position of the rolling element of the bearing after the discrete rolling element is shown in Figure 1a. The internal force distribution within the contact ellipse of the rolling element and aggregate element at the position shown in Figure 1a is solved to obtain the three-dimensional diagram of the internal force distribution, as shown in Figure 9 and Figure 10.

The internal force distribution in the contact ellipse at the bearing region and at the discrete groove is shown in Figure 9. The diagram shows that as the long semiaxis a and the short semiaxis b decrease, the corresponding internal force increases. The value of the internal force corresponding to the maximum internal force is located at the center of the ellipse. Figure 9a shows the internal force distribution at the center of the discrete groove. In this position, the long and short semiaxes of the contact ellipse are the shortest due to the position where the rolling element and aggregate element overlap the smallest amount, producing the smallest internal force between the two. Figure 9b shows the force distribution within the contact ellipse corresponding to rolling elements 2 and 3. Figure 9c shows the force distribution within the contact ellipse corresponding to rolling elements 4 and 5. Figure 9d shows the force distribution in the contact ellipse corresponding to rolling elements 6 and 7. A comparison of the three diagrams reveals that the closer the rolling element is to the discrete groove, the larger the corresponding ellipse size, and its maximum internal force is also the largest, which is consistent with the internal force variation law described in Figure 8.

As shown in Figure 11, the internal force distribution in the contact ellipse in the non- bearing region is the same as that in the bearing region and at the discrete groove. As the long and short semiaxes in the ellipse decrease, the internal force increases. Figure 10a shows the force distribution within the contact ellipse corresponding to rolling element 8 and rolling element 9. Figure 10b shows the force distribution within the contact ellipse corresponding to rolling element 10 and rolling element 11. Figure 10c shows the force distribution within the contact ellipse corresponding to rolling element 12 and rolling element 13. Figure 10d shows the force distribution in the contact ellipse corresponding to rolling element 14. A comparison of the above four diagrams shows that the closer the rolling element is to the uppermost part of the bearing, the smaller the internal force. The closer to the top of the bearing, the smaller the overlap between the rolling element and the aggregate element is, the smaller the internal force between the two, which also corresponds to the law depicted in Figure 8.

Finally, the internal forces exerted on the particles at different positions of the rolling body tumbling over the entire raceway are solved; their distribution pattern is shown in Figure 11.

According to the three-dimensional diagram in Figure 11, when the rolling element is on the groove, the internal force on the particle is the smallest. The internal force on the particle is the largest when the rolling element is on the bearing area, of which the closer the position on the bearing region is to the groove, the greater the internal force, which is consistent with the distribution law of the internal force between the rolling element and the aggregate element at different positions depicted in Figure 8. The closer the rolling element is to the center of the raceway and to the center of the contact ellipse, the greater the internal force on the particle in either position, which is consistent with the patterns depicted in Figure 9 and Figure 10.

#### 4.2.2. Force after Transmission

The internal force distribution law of each layer is obtained by solving the internal force transfer equation, as shown in Figure 12.

According to the theoretical analysis, Figure 12 shows that the internal force of the particles decreases as the number of layers increases. In each layer, the particles in the bearing region are subjected to the highest forces, and those in the discrete groove are subjected to the lowest forces. At any given position on the raceway, the central particle is subjected to the greatest force, which is consistent with the pattern depicted in Figure 10. This result indicates that cageless ball bearings are most severely worn in the bearing region and are least susceptible to wear in the discrete groove.

The particle arrangement in the main wear part of the bearing outer ring is hcp; only the bonding force between the two particles in the hcp arrangement is needed. According to the calculation of Equation (12), the maximum overlap *δ_p_* is taken, which is the particle radius, and the bonding force between the raceway particles in the initial state is 36.8858 N. When the force on the particle exceeds this value, the bond will break, and the bearing will wear. Analysis of the forces on each layer of particles shows that the values of the internal force s on some particles in layers 1 to 4 are greater than the bonding forces, and therefore, the depth of wear comprises four layers of particles, that is, 0.4 mm. The amount of bond breakage was then compared to the total number of bonds to obtain a wear rate of 8.67 × 10^−5^.

### 4.3. Experimental Results

After the wear test of the cageless ball bearing, the results of the test need to be analyzed in detail, mainly in terms of the amount of wear and the depth of wear of the bearing minus the bumper groove. In the test, the wear volume is determined using the weighing method. The bearing outer ring is weighed 10 times before and after wear, and the average value is taken as the reference data. The wear volume and wear rate of the outer ring of the bearing under different working conditions were calculated, as shown in Table 3.

As shown in Table 3, the wear of the outer ring of the bearing increases as the speed of the inner ring and the radial load increase, and the increase in speed has a greater effect on the wear of the outer ring than on the radial load. When the radial load is 2000 N and the inner ring speed is 6000 rpm, the wear rate of the bearing is 8.8158 × 10^−5^, which is within an acceptable margin of error of 1.65% from the theoretical calculation. This error can be attributed to errors in the weighing of the electronic balance and errors arising from the operation of the test.

As the wear amount determined by the weighing method can only determine the wear of the outer ring, the wear depth of the discrete groove is further measured using an ultrafield depth microscope. The three-dimensional measurement of the discrete groove under different operating conditions is shown in Figure 13.

The measured data were extracted and processed to obtain the wear depths of the individual specimens in the deconfliction grooves, as shown in Table 4.

As shown in Table 4, the depth of wear increases as the speed of the inner ring and the radial load increase. According to the wear depths given in Table 4, the wear depth of the grooves is 0.3651 mm for an inner ring speed of 6000 rpm and a radial load of 2000 N. This finding contradicts the theoretical analysis. The diameter of the particles in the theoretical analysis is 0.1; therefore, the resulting wear depths cannot be obtained to the exact percentile. However, both the theoretical wear depth and experimental wear depth lie between the 3rd layer and 4th layer of the particles and can therefore be verified against each other.

## 5. Conclusions

This paper combines the discrete element method with experiments to analyze the wear characteristics of discrete grooves in cageless ball bearings, with the following conclusions.

Using the discrete element method, the bearing discrete groove can be discretized into particles, and the particles are interconnected using bonds. Then, the wear of discrete groove was characterized by bond fracture. The feasibility of DEM in wear field is verified by theoretical and experimental analysis.Since the rolling element on the discrete groove does not carry and makes contact with two points, the contact force between the rolling body and the aggregate element is the smallest, the contact force with the bearing area of the raceway is the largest, and the contact force is greater with and closer to the discrete groove.Within the particle system, as the number of transfer layers increases, the internal force on the particles gradually decreases. In each layer of particles, the particles at the discrete groove are subjected to the least force, the particles in the load-bearing zone are subjected to the greatest force, and at any position on the raceway, the particles at the very center of the contact ellipse are subjected to the greatest force.The bond in the bearing discrete element model has bonding force. When the internal force transferred to the two particles under the external load is bigger than the initial bonding force, the bond will break. When the internal force on the particle is greater than 36.5558 N, the bond breaks, and the bearing wears out; otherwise, the bearing does not wear out.The wear of ball bearings without a cage increases as the speed of the inner ring and the radial load increase, and the influence of speed on wear is greater than that of the load. The most likely wear area is in the bearing area on the entire outer raceway and the least likely wear area is in the discrete grooves.

## Figures and Tables

**Figure 1 materials-15-06711-f001:**
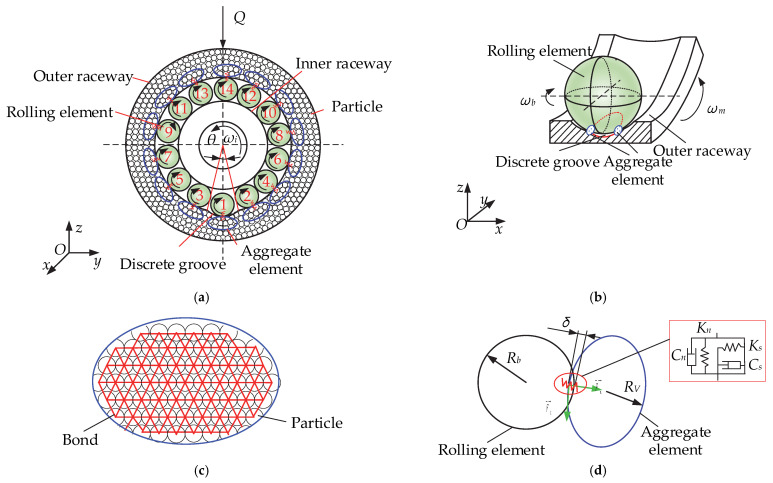
Discrete element model of cageless ball bearing: (**a**) discrete element model with rolling elements 1–14; (**b**) contact between rolling element and aggregate element; (**c**) aggregate element; (**d**) contact model of rolling element and aggregate element.

**Figure 2 materials-15-06711-f002:**
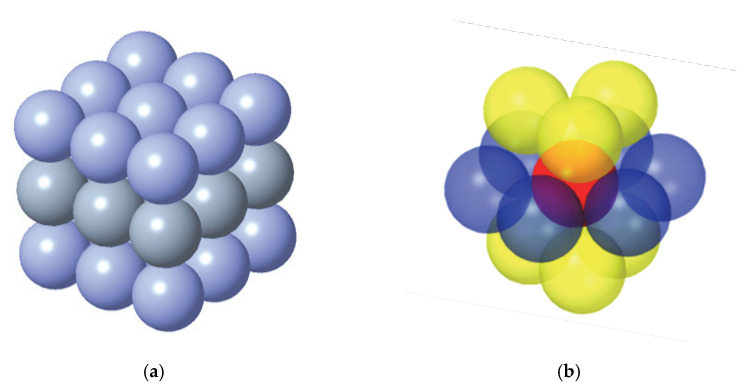
Particle arrangement: (**a**) simple cubic and (**b**) hexagonal close-packed.

**Figure 3 materials-15-06711-f003:**
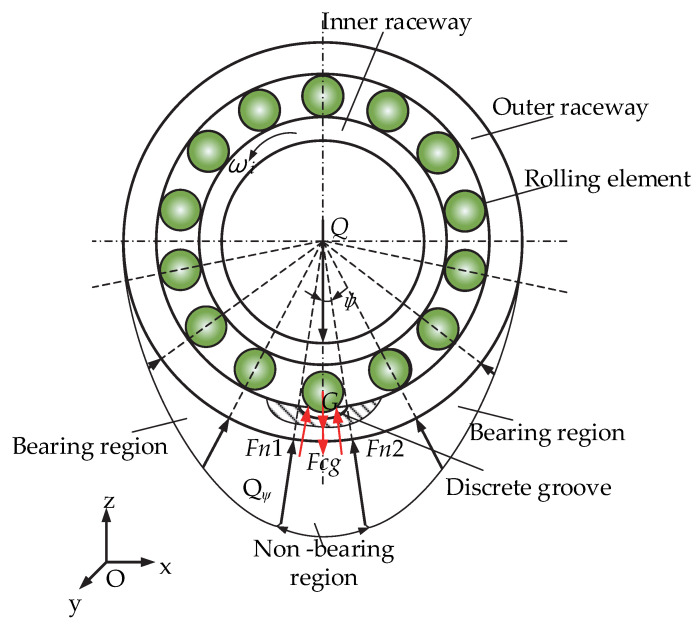
Force analysis of ball bearings without a cage.

**Figure 4 materials-15-06711-f004:**
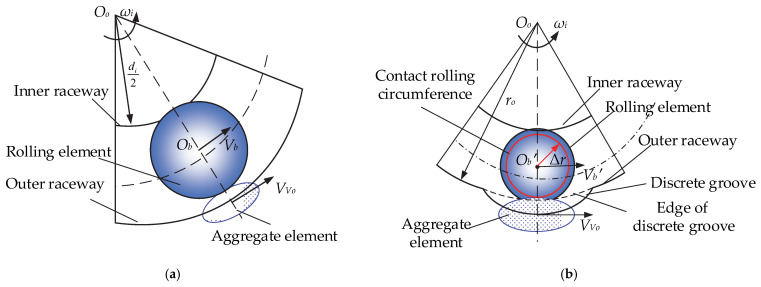
Analysis of the relative velocity of the rolling element to the aggregate element: (**a**) conventional raceway and (**b**) discrete groove.

**Figure 5 materials-15-06711-f005:**
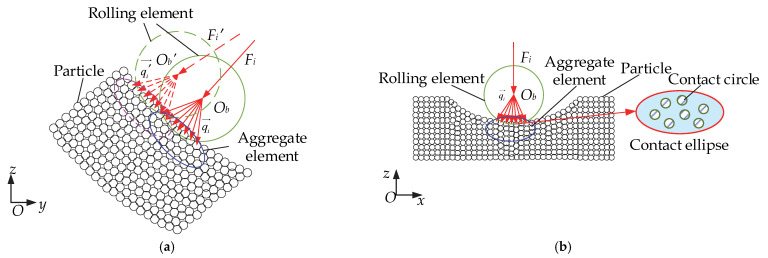
Rolling element in contact with the discrete outer raceway: (**a**) axial; (**b**) radial.

**Figure 6 materials-15-06711-f006:**
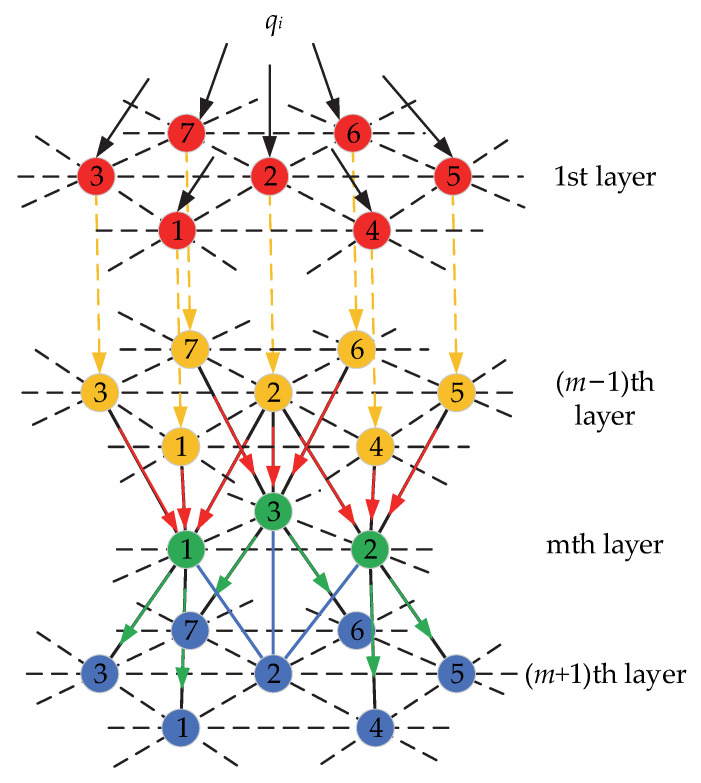
Hcp internal force transfer model.

**Figure 7 materials-15-06711-f007:**
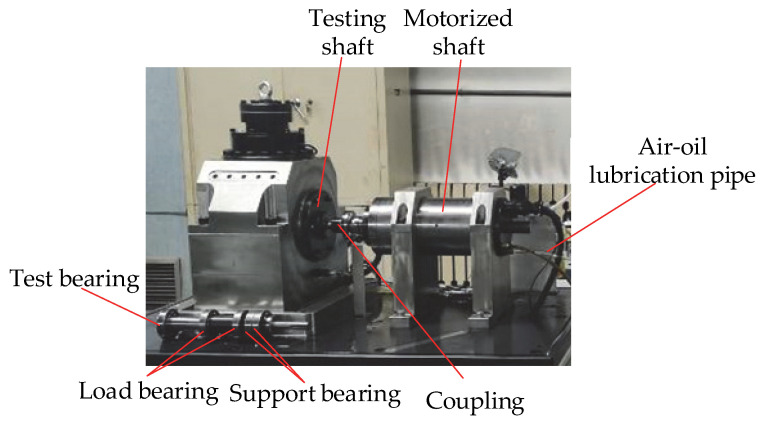
Experimental equipment.

**Figure 8 materials-15-06711-f008:**
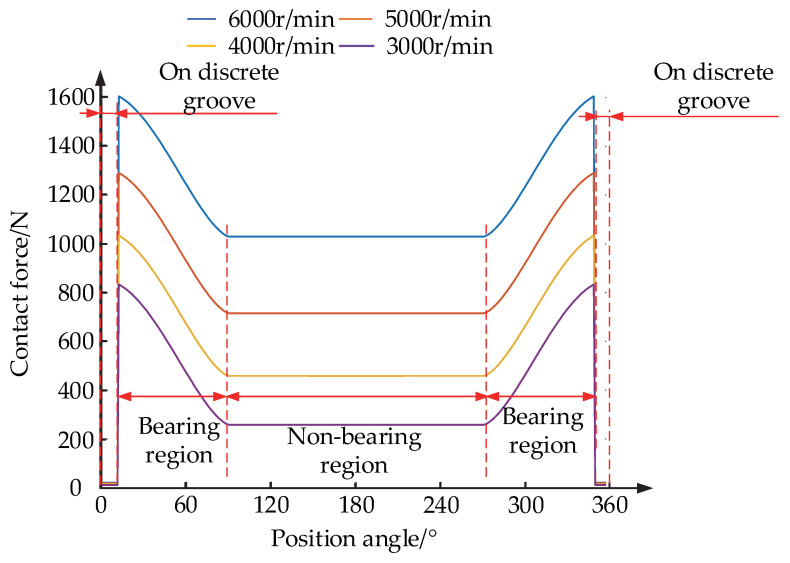
Variation in contact forces between the rolling element and the aggregate element at different positions of the aggregate element.

**Figure 9 materials-15-06711-f009:**
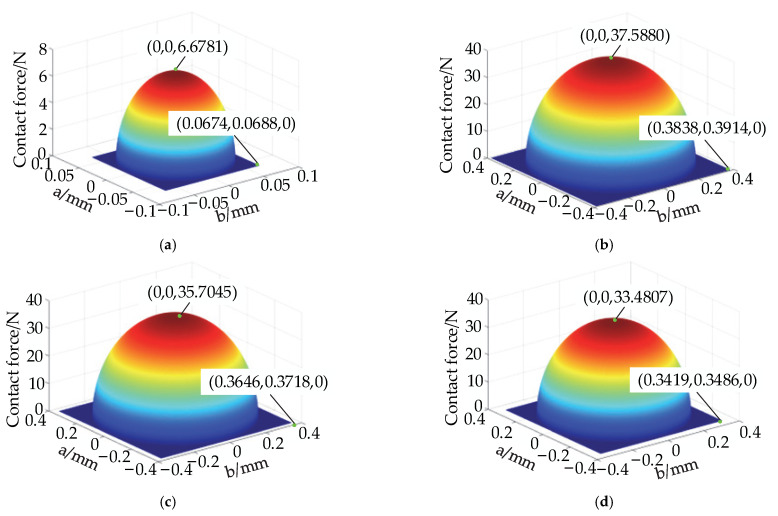
Internal force distribution in the contact ellipse at the bearing region and at the discrete groove: (**a**) rolling element 1; (**b**) rolling elements 2 and 3; (**c**) rolling elements 4 and 5; (**d**) rolling elements 6 and 7.

**Figure 10 materials-15-06711-f010:**
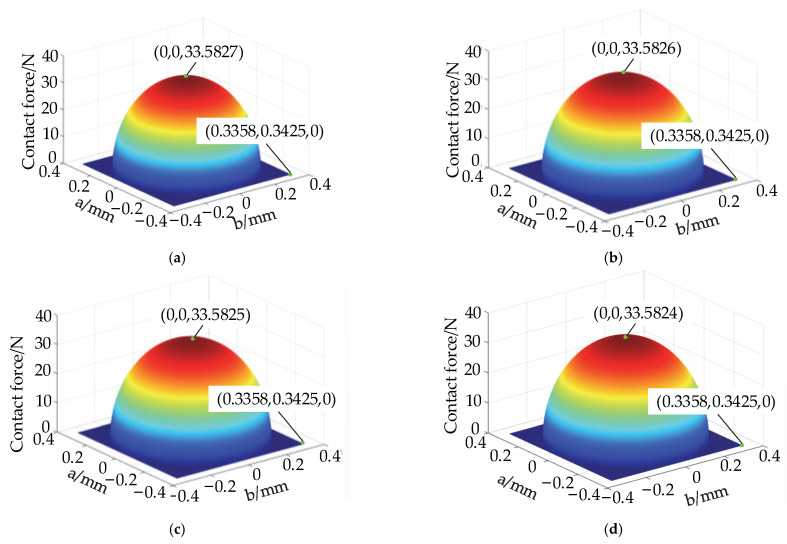
Internal force distribution in the contact ellipse in the non-bearing region: (**a**) rolling elements 8 and 9; (**b**) rolling elements 10 and 11; (**c**) rolling elements 12 and 13; (**d**) rolling element 14.

**Figure 11 materials-15-06711-f011:**
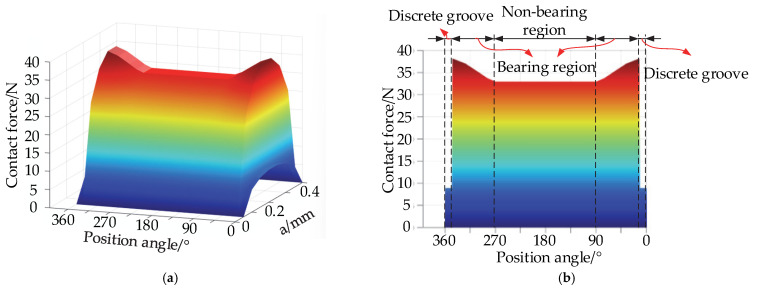
Internal force distribution pattern of raceway particles: (**a**) three-dimensional diagram; (**b**) main view.

**Figure 12 materials-15-06711-f012:**
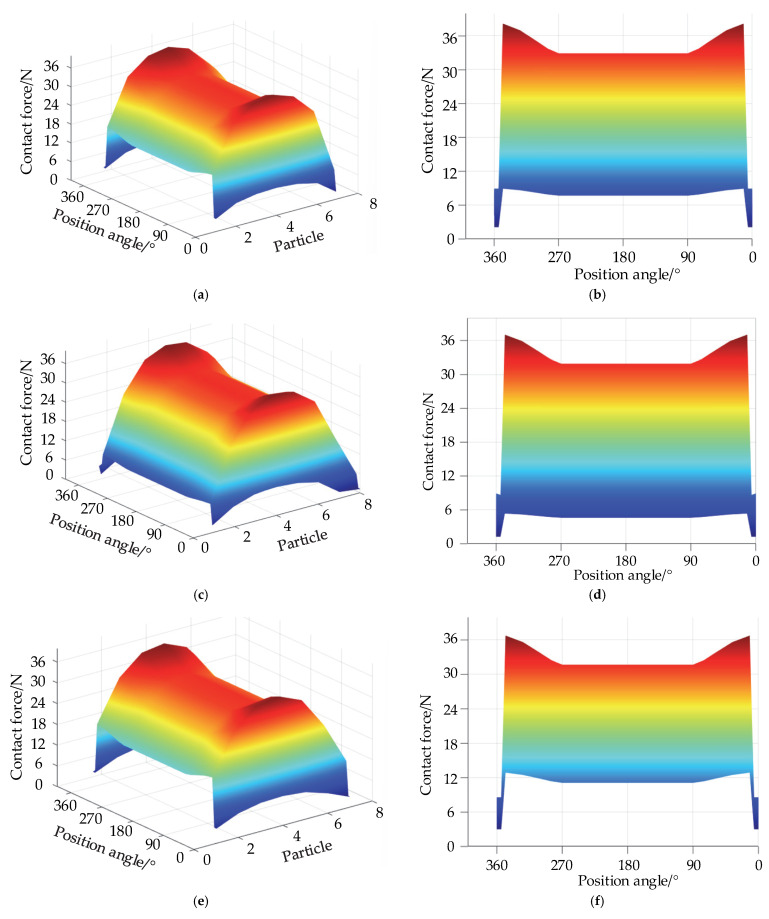
Internal force distribution of particles in each layer: (**a**) three-dimensional diagram of the second layer; (**b**) main view of the second layer; (**c**) three-dimensional diagram of the third layer; (**d**) main view of the third layer; (**e**) three-dimensional diagram of the fourth layer; (**f**) main view of the fourth layer; (**g**) three-dimensional diagram of the fifth layer; and (**h**) main view of the fifth layer.

**Figure 13 materials-15-06711-f013:**
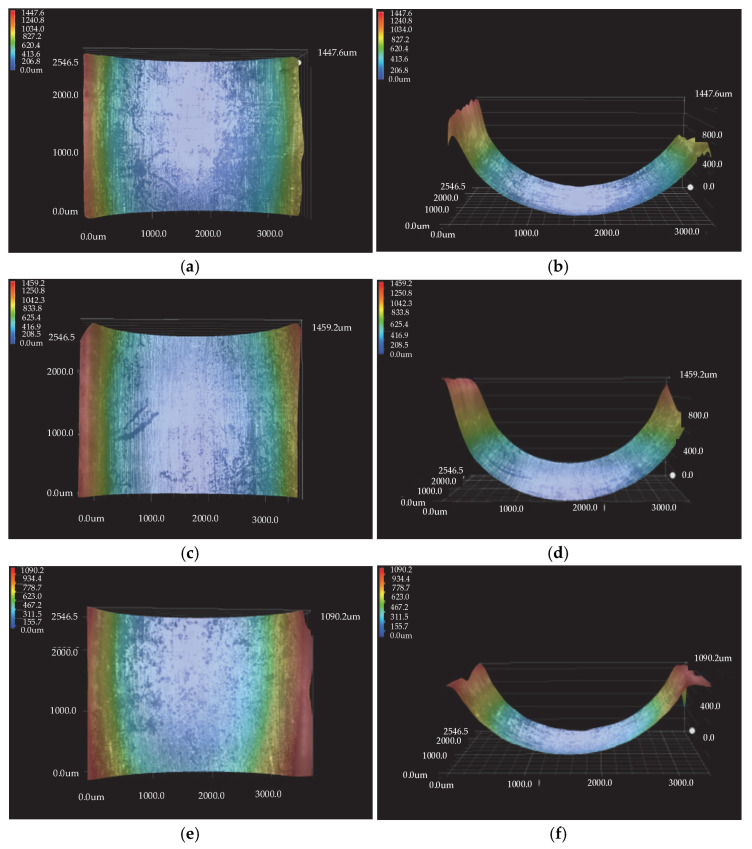
Three-dimensional measurement diagram of the discrete groove under different working conditions: (**a**,**b**) test piece 1; (**c**,**d**) test piece 2; (**e**,**f**) test piece 3; (**g**,**h**) test piece 4; (**i**,**j**) test piece 5.

**Table 1 materials-15-06711-t001:** Test plan.

Test Piece	Radial Load/N	Inner Raceway Speed/rpm	Experiment Time/h	Lubrication Status
1	2000	3000	3	dry friction
2	2000	4000	3	dry friction
3	2000	5000	3	dry friction
4	2000	6000	3	dry friction
5	2500	4000	3	dry friction
6	3000	4000	3	dry friction

**Table 2 materials-15-06711-t002:** Parameters for numerical solution of contact forces for rotational speed variations.

Parameters	Numerical Values
Inner raceway speed *ω_i_*/(r/min)	3000~6000

**Table 3 materials-15-06711-t003:** Wear rate.

Test Piece	Mass before Wear/g	Mass after Wear/g	Wear Weight/mg	Wear Rate
1	97.8781	97.8761	2	2.0434 × 10^−5^
2	97.7477	97.7445	3.21	3.284 × 10^−5^
3	97.0451	97.0383	6.82	7.0277 × 10^−5^
4	97.6655	97.6569	8.61	8.8158 × 10^−5^
5	97.6569	97.6529	3.98	4.0755 × 10^−5^
6	97.0384	97.0337	4.67	4.8125 × 10^−5^

**Table 4 materials-15-06711-t004:** Wear depth.

Test Piece	Wear Depth
1	0.3011
2	0.30236
3	0.3547
4	0.3651
5	0.33547

## Data Availability

Please contact the corresponding authors.

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
