# Peer review of "Characterization of Bond Fracture in Discrete Groove Wear of Cageless Ball Bearings"

_materials, 2022, doi:10.3390/ma15196711_

Round 1

Reviewer 1 Report

 I would like to inform you that the subject is relevant and interesting. The manuscript adds new information compared with other published material. The language of the paper is quite good and the paper is well written and the text is clear and easy to read. In addition to that, the conclusions are consistent with the evidence and arguments presented. 

In table 3, it is preferable to replace the term of mass by weight. 

Reviewer 2 Report

Very interesting article, very well written, so it may be of interest to potential readers. Introduction correctly written. The authors referred to the research of other scientists. I have no comments to the introduction. The test methods are described in some detail. This chapter can be significantly shortened. I haven't noticed any information about the research standards. There is no explanation on what basis the scientists chose the rotational speed, load and test time. The analysis of the presented tests was carried out correctly. I propose to improve the summary and draw more detailed conclusions. Literature appropriate to the content of the article.

Reviewer 3 Report

This paper studies the discrete groove wear of 2 cageless ball bearings and corresponding bond fracture.

The paper is well organized. Abstract, introduction, experimental procedure and results and discussion sections are well structured and contain enough information. Results are reliable and contain useful information.

I want to recommend the publication of the paper. Just I have minor comments as followings:

- the symbols in Fig. 1 must be described in text or at least in the figure caption. This is applicable for all figures.

- it would be better to include a separate section and “results” after experimental procedure and before discussion section. Or authors can include a section as results and discussion. This may help the reader to follow the manuscript more easily.
